# Butyric and Citric Acids and Their Salts in Poultry Nutrition: Effects on Gut Health and Intestinal Microbiota

**DOI:** 10.3390/ijms221910392

**Published:** 2021-09-27

**Authors:** Mebratu Melaku, Ruqing Zhong, Hui Han, Fan Wan, Bao Yi, Hongfu Zhang

**Affiliations:** 1State Key Laboratory of Animal Nutrition, Institute of Animal Science, Chinese Academy of Agricultural Sciences, Beijing 100193, China; sefibahir2009@gmail.com (M.M.); zhongruqing@caas.cn (R.Z.); hanhui16@mails.ucas.ac.cn (H.H.); wanfanfw@126.com (F.W.); 2Department of Animal Production and Technology, College of Agriculture, Woldia University, Woldia P.O. Box 400, Ethiopia; 3College of Pastoral Agriculture Science and Technology, Lanzhou University, Lanzhou 730020, China

**Keywords:** butyric acid/salt, citric acid/salt, gut health, intestinal microbiota, poultry nutrition

## Abstract

Intestinal dysfunction of farm animals, such as intestinal inflammation and altered gut microbiota, is the critical problem affecting animal welfare, performance and farm profitability. China has prohibited the use of antibiotics to improve feed efficiency and growth performance for farm animals, including poultry, in 2020. With the advantages of maintaining gut homeostasis, enhancing digestion, and absorption and modulating gut microbiota, organic acids are regarded as promising antibiotic alternatives. Butyric and citric acids as presentative organic acids positively impact growth performance, welfare, and intestinal health of livestock mainly by reducing pathogenic bacteria and maintaining the gastrointestinal tract (GIT) pH. This review summarizes the discovery of butyric acid (BA), citric acid (CA) and their salt forms, molecular structure and properties, metabolism, biological functions and their applications in poultry nutrition. The research findings about BA, CA and their salts on rats, pigs and humans are also briefly reviewed. Therefore, this review will fill the knowledge gaps of the scientific community and may be of great interest for poultry nutritionists, researchers and feed manufacturers about these two weak organic acids and their effects on intestinal health and gut microbiota community, with the hope of providing safe, healthy and nutrient-rich poultry products to consumers.

## 1. Introduction

Organic acids are weak acids having a carboxylic acid group (R-COOH), intermediates in the degradation pathways of carbohydrates, amino acids and fats, and are used as nutritional value and antimicrobial effects in animal feeds [1,2,3].

The use of organic acids in animal feeds started many years ago due to the ban on the use of antibiotics [4]. They are used as an antibiotic alternative that can alter the physiology and lead to the death of pathogenic microorganisms in animals, including poultry [5,6]. Various literatures reveal that antibiotics have better positive effects in modulating metabolism, improving weight gain, feed efficiency and controlling diseases in poultry production [7,8]. However, their continued use in animal nutrition developed antibiotic resistance and drug residues, which resulted in global public health issues and exacerbating poverty in the 21st century [9,10,11]. In this regard, organic acids are selected as a promising feed additive in poultry production due to their ability to maintain gut barrier cellular integrity, modulate intestinal microbiota, improve digestion and nutrient absorption rate and contribute to improved production performance [6,12,13]. BA as a short chain fatty acid (SCFA) and CA as a tricarboxylic acid (TCA) gained considerable attention as representative organic acids in poultry production. They are used as an energy source of prime enterocytes [14] or for the bactericidal efficacy against harmful species (for example, *Escherichia coli*) and the enhanced bone mineralization and improved function of gut microorganisms [15,16,17]. As organic acids, they are volatile and corrosive in their free forms; thus, they are commercially produced into salt forms [18,19,20] to increase palatability and bioavailability in the gut of birds [21,22]. Previous studies revealed that addition of active ingredients in salt forms into the diet of monogastric and young ruminants [16,23] could improve gut microbiota diversity and intestinal health and reduce microbial infections [24].

Each organic acid has its specific ability against pathogenic bacteria. For example, compared to medium-chain fatty acids (MCFAs), butyrate has a less strong anti-bacterial effect, although it has been widely used in animal production because of its low price [25,26,27]. Furthermore, previous studies largely focused on supplementation of blends of organic acids within short-chain fatty acids (SCFAs) or SCFAs with MCFAs on poultry challenged with *Clostridium perfringens* [28], *Eimeria* spp. [27] and *salmonella typhimurium*-related diseases [29,30]. A meta-analytic study of organic acids confirmed that blends of two or more specific organic acids improved performance, immunity and welfare of birds better than any acid achieved alone [4,8]. For example, Ndelekwute et al. studied the effects of four different acidifiers on gut performance and found that organic acids could be used in diets for broilers [31]. However, studies on blends of BA and CA on performance, nutrient digestion, intestinal health and meat quality of birds have not been found. They can be of great research interest for animal nutritionists and researchers. In addition, both have different modes of action and commonly increase the acidity of gut digesta and are important to keep the digestible nutrients under normal physiological conditions. Therefore, this review briefly summarizes the discovery of BA, CA and their salt forms, molecular structure and properties, metabolism, biological functions and their applications in poultry nutrition.

## 2. Discovery, Molecular Structure and Properties of BA and CA

BA, a SCFA with a four-carbon (C4) chain length was discovered by Adolf Lieben and Antonio Rossi in 1869. The name BA came from the Latin word, *butyrum* or *buturum,* meaning the acid of butter, as it was discovered from rancid butter [32,33]. It has a molecular formula of *C_4_H_8_O_2_* and structural formula of *CH_3_CH_2_CH_2_COOH* [34]. It has synonyms called *butanoic acid* (*CH_3_CH_2_CH_2_CO_2_H*), *n-butyric acid* (a substance that was isolated from butter in 1869) and *n-butanoic acid* (International Union of Pure and Applied Chemistry, IUPAC) [35]. BA has a melting point of −7.9 °C, boiling point of 163.5 °C [36], molecular weight of 88.11 g/mol [37] and pKa value of 4.82 [38]. Butyrate and butanoate are also salts and ester forms of BA, respectively [39]. BA has an unpleasant odor, is a colorless liquid, and is potentially volatile and soluble in water, ethanol, and ether property [40]. Naturally, BA is synthesized from dietary fibers by anaerobic bacterial fermentation in the gut of mammals and birds [41,42].

Similarly, CA is a TCA or Krebs cycle acid with a six-carbon (C6) chain length discovered by Swedish chemist Carl Wilhelm Scheele in 1784 by crystallizing it from lemon juice [43,44]. The name CA came from the Latin word *citrus*, a tree naturally derived from citrus fruits and juices [45]. *3-*carboxyl and *1-*hydroxyl groups present in CA were recognized by Liebig in 1838, and calcium citrate was prepared from CA in 1860 in the United Kingdom, and in 1880 in France, Germany and the United States of America [46]. It is a weak organic acid with a chemical formula C_6_H_8_O_7_ and IUPAC name *2-*hydroxypropane*-1,2,3-*tricarboxylic acid, also known as *β-*hydroxy-tricarballylic acid [47]. It has a boiling point of 175 °C, melting point of 153 °C and density of 1.67 g/cm^3^ [48] and molar mass of 192.12 g/mol [49]. In addition, CA has a molecular weight of 210.14 g/mol, gross energy 10.3 KJ/g [50] and three pKa values (pKa1 = 3.1, pKa2 = 4.7 and pKa3 = 6.4) [51]. It is an odorless, colorless crystal, highly soluble in water, ethanol and a sour taste property [52]. CA is normally used as a feed acidifier, flavoring agent and preservative in foods, beverages, detergents, cosmetics, toiletries and pharmaceuticals [53]. CA is a normal constituent in human and animal diets [54] and an intermediary substance in oxidative metabolism [55]. It is quickly metabolized to CO_2_ and H_2_O after ingestion. Supplementing CA in animal feed is safe and poses no risk to the environment [56]. Recently, the demand for commercially produced BA and CA acidifiers have increased worldwide [57,58,59].

## 3. Metabolism of BA and CA in Poultry

BA, CA and their salt forms play a crucial part in energy metabolism and keep gut homeostasis and epithelial integrity, participating in immune response, suppressing inflammation, and reducing oxidative stress in farm animals, such as poultry and pigs, and humans [60,61,62,63,64,65,66]. When free BA is given orally to poultry, it is rapidly metabolized and absorbed in the crop’s mucosa in the acidic environment of the gizzard and proventriculus [67]. This results in a higher concentration of BA in the foregut and leads to higher proteolytic activity of birds [68]. Therefore, protected butyrate is produced and fully utilized in the colon and cecum, which helps to improve epithelial barrier function, reduce inflammation and limit the invasion of pathogenic bacteria [67,69,70]. Butyrate is also mainly produced from dietary fibers (such as, cereals and grains) to a lesser extent from proteins via bacterial fermentation in the colon of mammals and the cecum of chicken [71,72,73,74]. *Bacteroidetes* and *Firmicutes* phyla are the most dominant chicken cecum microbiota (80%). They degrade structural carbohydrates and specific soluble oligosaccharides [69,75] that escape in the upper part of the digestion process [76,77]. In the cecum of chicken, bacterial fermentation coverts dietary fiber (complex polysaccharides) to monosaccharides and then to pyruvate and acetyl-CoA by pentose phosphate and glycolytic pathways [77]. Butyrate, the main energy source, is formed from acetyl-CoA condensation and a stepwise reduction of butyryl-CoA by two metabolic pathways [78] (Figure 1). The first pathway, Butyryl-CoA, an intermediate for the four-step pathway of butyrate production, is transformed to butyryl-phosphate via phosphorylation by the enzyme phospho-transbutyrylase. Butyryl phosphate is then converted to butyrate by the butyrate kinase enzyme [79,80]. In the second pathway, the enzyme butyryl-CoA: acetate CoA transferase, found in most gut bacteria families, transfers the acetyl-CoA moiety of butyryl-CoA to external acetate, leading to the formation of butyrate and acetyl-CoA [81]. Then, butyrate is absorbed in the gut lumen by enterocytes through two mechanisms as described by [82]. First, through simple diffusion of the undissociated form [42], which is used for villus growth and cell turnover. Second, in dissociated form, which is activated by the SCFA transporters such as monocarboxylate transport isoform 1 (MCT1), a H^+^ coupled transporter and sodium-linked monocarboxylate transport 1 (SMCT1). SMCT1 also known as solute carrier family five-member-eight (SLC5A8), a Na^+^ coupled transporter found only in the apical membrane of colonic epithelial cells [83]. Orally given butyrate is more efficient and quickly absorbed than naturally produced by bacterial fermentation in the cecum of birds [84]. The type of BA, form and salt or protection structure also affects its absorption rate in the gut. The addition of soluble fibers in the diet [85] and the addition of exogenous enzymes such as xylanase, which converts dietary arabinoxylans (main non-starch carbohydrates in wheat) into xylo-oligosaccharides [86,87], are also the most important strategies to increase BA availability and absorption in the GIT of birds.

Similarly, CA is the first intermediate metabolic product formed through the TCA cycle [88] and a crucial element in the metabolic conversion of carbohydrates, fats and proteins to CO_2_ and H_2_O [33]. CA is produced by the reaction of oxaloacetate and acetyl-coenzyme A to yield citrate through citrate synthase enzyme [89,90]. In the mitochondria, acetyl-CoA is converted from pyruvate and used for energy production, whereas oxaloacetate is produced from pyruvate and CO_2_ using the enzyme pyruvate carboxylase in the cytoplasm [91].

Pyruvate is then transferred into the mitochondria and converted to acetyl-CoA by releasing CO_2_. Energy is then released and captured in the TCA cycle in the form of NADH, FADH_2_ and ATP [92,93]. Wolffram et al. and Tugnoli et al. summarized the study of intestinal absorption of CA from pig proximal jejunum [94,95], which may provide a reference for poultry. CA is absorbed via a Na-dependent co-transporter on the apical side and metabolized in the enterocyte directly influencing piglet’s intestinal metabolic status.

Literature from previous human studies also reveals that citrate is absorbed from the diet in the small intestine by means of the Na^+^-dicarboxylate cotransporters (NaDC_1_ and NaDC_2_) [96,97,98]. The enterocytes of the small intestine transport citrate out of the cells in the intestinal lumen. When CA is taken in by animals, it usually develops into the form of CA salts in the body. In general, there are limited findings on CA salts such as citrate (Na or K); metabolism in farm animals and human studies can be a baseline for future research work in poultry.

## 4. Biological Functions of BA and CA

### 4.1. Antibacterial Function, Acidity, Nutrient Absorption and Performance

Campylobacter sp., *Salmonella* Typhimurium, *Escherichia coli* sp., *Shigella* sp., *Clostridium perfringens* sp. and other Gram-negative bacteria species are common pathogens found in poultry farming [99,100]. These pathogens damage the villus–crypt units and intestinal mucosa of birds, lower the surface area for nutrient digestion and absorption, then reduce overall performance [101,102]. BA and CA are weak organic acids having different antibacterial actions, depending on the pKa value of the acids and intestinal pH [38,103] that cause the death of harmful bacteria in the gut of animals [16,104]. In BA, the undissociated form will enter and dissociate into butyrate (CH_3_CH_2_CH_2_COO^−^) and release H^+^ ions inside pathogenic bacteria cytoplasm [105,106]. This phenomenon lowers the pH value in the stomach/gut of livestock, causing enzymes inactivation [14], destroys DNA replication abnormity, and then disrupts pathogenic bacteria’s normal metabolic function [107]. Besides, the pathogenic bacteria itself consumes energy by activating proton pumps to fight against the lowering of the acid in the cell wall, thus inhibiting pathogenic bacteria growth and colonization in poultry intestines [16], whereas CA’s antibacterial activity creates an acidic condition in the stomach (pH 3.5–4.0), which prevents the growth of *Salmonella*, *Escherichia coli* and other acid-intolerant Gram-negative bacteria in the GIT of birds [108]. Its mechanism is to work through the activation of proteolytic enzymes and decrease the risk of sub-clinical infection [17].

Similarly, the acidic environment produced by CA in the stomach promotes *lactobacilli* growth and prevents pathogenic bacteria multiplications [109]. In addition to the antibacterial function of BA and CA, various studies examined their effects at different dose levels, product forms, and comparisons with antibiotics on nutrient digestibility, growth, and meat yield of poultry. Broilers fed at 0.2% BA significantly increased carcass weight and breast meat yield versus birds fed the control diet [110]. A supportive study on sodium butyrate (SB) at 0.6 and 1.2 g/kg in a broiler significantly increased average daily gain (ADG) (27.6 g) and feed conversion ratio (FCR) (1.8) during 1–21 days [111]. Salt forms of BA are slowly absorbed in the foregut gut (crop, proventriculus and gizzard) and are more effectively absorbed in the hindgut (duodenum, jejunum, ileum and ceca), inhibit the growth of pathogenic bacteria, and they improve the overall performance of birds [22,38]. Song et al. also proved the effects of microencapsulated SB (MESB) orally infected or uninfected with Eimeria species and *Clostridium perfringens* at 12 days of age followed by an oral inoculation with *Clostridium perfringens* at 16, 17 and 18 d of age on the growth performance of broilers. Broiler-fed MESB at 800 mg/kg feed challenged with necrotic enteritis showed higher total body weight, daily gain, and FCR at 35 days [112]. In contrast, insignificant changes observed in growth performances of commercial laying hens supplemented with 193, 136 and 198 g/t protected SB (PSB), respectively, whereas the quadratic effect showed maximization of eggshell thickness, percentage and strength sequentially with addition of PSB at 112 days of age [113]. Similarly, the antimicrobial effects of *n-butyric acid* and its derivatives (Monobutyrin (MB) and a mixture of mono-, di-, and tri-glycerides of BA) at concentrations from 250–7000 mg/kg inoculated either Salmonella typhimurium or Clostridium perfringens. The results showed that *n-butyric acid* and 50% MB could be used to control Salmonella Typhimurium or Clostridium perfringens in poultry [114].

Similarly, protected calcium butyrate (PCB) feed for broilers was at 0, 0.2, 0.3 and 0.4 g/kg and the results showed chicken-fed PCB at 0.3 g/kg had higher weight gain (125 g) than 0.2 (80) and 0.4 g/kg (83 g), respectively. Moreover, the apparent overall crude fat digestibility, apparent nitrogen corrected metabolic energy and FCR, increased during the entire experimental period [67]. The reason may be that butyrate increases the cell concentration of Ca^2+^ pancreatic cells, inhibits the growth of bile salt deconjugating bacteria, reduces the utilization of nutrients by microorganisms, and improves the digestibility and absorption rate of nutrients in broilers. Comparatively, Chowdhury et al. examined the effects of CA at 0.5% and avilamycin at 0.001% on broilers and obtained significant growth performance parameters versus avilamycin or control diets at 35 days of age [108]. Similar CA findings in laying hens and broilers showed a positive response to stimulate pepsin activity, support protein digestion, increase apparent digestibility and phosphorus bioavailability [115,116].

In summary, several studies recommended the importance of BA, CA and their salts to reduce the load of pathogenic microorganisms in the intestine, activate digestive enzymes, improve the digestibility and absorption of nutrients, gut microflora function and performance of birds [105,113,117,118,119]. Table 1 also summarizes various findings that support the above summary.

### 4.2. Gut Morphology and Barrier Function

Gut morphology, barrier function, and intestinal microbiota community of birds are vulnerable due to many detrimental environmental or nutritional factors [100], which lead to leaky gut, dysbiosis, failed intestinal barrier permeability and intestinal inflammation in poultry [127,128]. The intestinal dysfunction may cause reduced nutrient absorption surface area and growth performance [129]. Supplementation of BA, CA and their salts are among the strategies to modulate gut microbiota and keep poultry intestinal health [130]. They can promote intestinal epithelial cell proliferation and increase villus height (VH), thereby improving the absorptive surface area of the GIT [107]. Broilers supplemented with coated SB (CSB) at 0, 200, 400, 800 or 1000 mg/kg showed improved intestinal integrity by stimulating goblet cells in jejunum and increasing ileal VH at 42 days age [22]. The authors also found that SB at 800 mg/kg can lead to higher total antioxidant capacity (T-AOC) and reduced malondialdehyde (MDA) content in chicken jejunal mucosa. Likewise, Elnesr et al. mentioned the inclusion of SB at 0.5 and 1 g/kg increased villus length (VL) at day 21 (55 and 27%) and day 42 (39 and 18%), respectively, versus the basal diet [106]. Butyrate can also benefit pro-inflammatory cytokines such as tumor necrosis factor α (TNF-α), interleukin-1 (IL-1), interleukin-2 (IL-2), and interleukin-6 (IL-6) which are known to increase epithelial cell permeability in young broiler chicks [131]. Zou et al. conducted confirmatory studies to prove the above statement by supplementing SB at T1: CON, T2: DSS, T3: 150 mg/kg SB and T4: 300 mg/kg levels in female Chinese Yellow broilers. The result revealed 300 mg/kg SB significantly reduced IL-6 and IL-1β levels, whereas it increased IL-10. At the same time, it reduced the lesion score of intestinal bleeding and increased VH and the total mucosa area of the ileum [132]. These cytokines cause a homeorhetic response that changes the portioning of nutrients during inflammatory reactions in the gut of chickens [133]. In other studies, calcium butyrate administration in the colon of rats and BA in humans was found to have anti-inflammatory effects of treating inflammatory colon diseases [134,135]. Yan and Ajuwon also confirmed that butyrate could decrease lipopolysaccharide (LPS) damage on intestinal barrier integrity and tight junction (TJ) permeability and increased the abundance of claudin-3/4 expression [136]. MESB upregulated TJ protein expressions such as claudin-1, claudin-4, occludin, ZO-1, mucin-2, chicken liver-expressed antimicrobial peptides (cLEAP-2), and thus reduced intestinal mucosal barrier damage in *Necrotic enteritis* infected broilers [112]. A recent study also reported similar positive results on intestine gene expression in *coccidia* infected broilers supplemented with tributyrin (TB) [137]. The better secretion of mucin-2 by goblet cells prevents the attachment of pathogens to the epithelial tissues by cLEAP-2. Many researchers studied anti-inflammatory and immune-enhancing properties of SB, which influences IL-6, IL-8, IFN-γ, TGF-β and IL-1β inflammatory cytokine expression in broilers and piglets [22,138,139].

CA causes an acidic condition in poultry gut, decreases pathogenic bacteria and improves intestinal morphology and barrier function [140]. CA administration at 3% and 6% significantly increased VL, crypt depth (CD) and goblet cell numbers in the duodenum, jejunum and ileum as well as villus weight (VW) and villus length to crypt depth (VL: CD) ratio in the duodenum of broilers at 42 days of age [141]. A study by Nourmohammadi and Khosravinia also revealed that 30 and 60 g/kg CA in broiler diets significantly improved the weight of proventriculus, gizzard, ileum and length of jejunum and ileum as compared to the control [142]. A similar finding by Khosravinia et al. [126] proved a significant increment in VL, CD and the number of goblet cells in the small intestine of broilers fed 30 g/kg CA with corn-soybean diets. CA decreases pathogenic bacteria colonization and limits toxic metabolite production in the GIT of birds [143]. In general, BA and CA with their salts can be a potential feed additive to maintain gut integrity, improving barrier function and enhancing poultry productivity (Table 2).

### 4.3. Immune Function and Antioxidation

Additionally, the crucial role of dietary BA and CAs in enhancing the immune system function was mentioned by various authors [16,65,144]. Butyrate also influences the immune response of animals by affecting immune cell migration, adhesion, proliferation and differentiation [60,80], and maintains gut homeostasis in chickens [145]. Many studies suggested that BA as a feed additive for animals decreased oxidative stress and contributed to better nutrient digestibility and growth rate [4,57,146]. It is proven that supplementation of butyrate glyceride can modulate intestinal microflora and serum metabolites to maintain intestinal metabolism homeostasis in broilers [76]. Recent studies showed linear inclusion of SB significantly increased the relative weight of the thymus, the foundation to achieve immune function and antibody titer, which improves the humoral immunity of broilers, protecting against *Newcastle* disease (NCD) infection [111,147]. Butyrate has powerful effects on several colonic mucous functions, such as inflammatory suppression and carcinogenesis, improving colonic protective barrier elements and reducing oxidative stress [82]. Orally administered butyrate at 0.25 or 1.25 g/kg doses for 5 days caused histone protein H2A hyperacetylation in broilers regardless of the dosage level [148].

Similarly, Moquet found that butyrate supplementation can raise the concentration of digesta butyrate in the gastric, ileum and colon regions. It helped β-oxidation of lipids and showed a positive immune response in chickens [131]. The reason may be the promotion of glycolysis and location impact of butyrate on energy metabolism in the GIT of broilers. Zhou et al. also studied immunomodulatory and protective effects of butyrate on the avian macrophages in the presence or absence of LPS challenged by *salmonella typhimurium*. The results showed that butyrate inhibited IL-1, IL-6, and IFN-γ expression in LPS-stimulated cells, suggesting that it could be used to regulate inflammation and immune homeostasis in chickens [29]. This is due to the inhibitory action of butyrate on the activation of nuclear factor (NF-κB) via IκB-α and IκB-β stabilization.

Furthermore, Mátis et al. examined the effects of orally given sodium butyrate at 0.25 g/kg body weight on insulin signaling of broilers from 20–24 days. Orally butyrate-treated groups increased glucose plasma and insulin levels versus the control birds [72]. Since oral butyrate administration is more effective and absorbed faster [148], it appears to act as a bioactive molecule in extrahepatic tissues, causing changes in insulin signaling. Besides, butyrate also acts as a ligand for GPCRs such as GPR109A, GPR43, and GPR41, contributing to the immune system, homeostasis, and inflammation by activating anti-inflammatory signaling cascades [83,149].

CA also enhances the density of lymphocytes, a principal constituent of the bird’s immune system, to combat antigens in the lymphoid organs and boost non-specific immunity [107]. CA at 0.5% improved specific and non-specific immunity against NCD vaccinated broilers [150]. Furthermore, the inclusion of CA at 0.5% in a corn–soybean basal starter chicken diet improved tibial ash deposition, lymphocyte organs, and tissue densities to fight against pathogens [151]. Besides, supportive findings showed that broilers supplemented with 30 and 60 g/kg CA increased thymus and bursa fabricius index, respectively, which correlates with the improved immune response of broilers at 42 days [142]. The authors concluded that 60 g/kg CA caused acidic stress, severely reduced performance and disrupted liver function in chickens.

Conversely, Lakshmi and Sunder recommended that 2% CA was the ideal inclusion level and more effective in stimulating humoral immune response and higher antibody titers in broilers at 42 days of age [152]. Similarly, previous research findings stated that the inclusion of CA at 2.5% in the diet of rabbits improved lymphocyte cells, which are the second most common leukocytes that increase the protection mechanism against non-specific pathogens to boost immunity [153]. Based on the various research findings, it can be concluded that the relative increment in the weight of immune organs, inactivation of NF-κB with dietary supplementation of BA or CA with their derivatives improved immune responses and health of broilers (Table 2).

**Table 2 ijms-22-10392-t002:** The responses of different forms of BA and CA on histomorphology, immune organs and serum biochemistry of broilers.

Forms of BA/CA	Broiler strain and Trial Duration (Day)	Study Layout and Dosage Levels	Responses Expressed as a Percentage of Respective Controls	Reference
BA	Broiler chicks for 42 days	T1: CTR, T2: 20 mg/kg BMD, T3: 3 g/kg BA, T4: 4 g/kg BA	↑ GLUT5, SGLT1 and PepT1 expression. ↑ humoral, cell-mediated immune responses and serum biochemistry at T4. ↑VL and VD.	[154]
SB	M77 Hubbard broiler at d-21 and d-35	T1: CTR, T2: 0.1 g/kg ZnB, T3: 0.5 g/kg SB, T4: 1.0 g/kg SB	↑ antibody titer against NCD and SRBCs. ↑ Thymus, spleen and bursa weight. ↑ Duodenum and Jejunum VH. ↑Goblet cells in the SI and ileum.	[147]
SB	Cobb 400 broiler for 42 days	T1: CTR, T2: AB (50 ppm), T3: 0.09% CSB, T4: 0.18% CSB, T5: 0.03% UCSB, T6: 0.06% UCSB	Cecal *Escherichia coli* and *Clostridium perfringens* count reduced with the addition of CSB. ↑ Jejunum VH, VH: CD ratio and VH: VW ratio with addition of CSB by 0.18%.	[155]
SB	Arbor Acres broilers for 45 days	T0: CTR, T1: 0.3, T2: 0.6 and T3: 1.2 g/kg SB	↑ weight and length of duodenum, jejunum, ileum, SI, pancreas, thymus, and length of caeca. ↑ Antibody titer against NCD.	[111]
SB	Broiler chicks	SB with or without *Salmonella typhimurium* (LPS) challenged disease	SB ↓IL-1, IL-6, IFN-γ, and IL-10 in LPS-stimulated cells. ↓TGF-3 expression in both cases.	[29]
ESB	Female Chinese Yellow broilers	T1: CON, T2: DSS, T3: 150 mg/kg SB, T4: 300 mg/kg SB	↓ Lesion scores of intestinal bleedings. ↑ VH and ileum total mucosa. ↓ D (-)-lactate level, IL-6, and IL-1β. ↑ interleukin-10.	[132]
PPSB	Mixed Cobb chicks at 1–14, 15–28 and 29–42 days.	T1: CTR, T2: AB (100,000 IU/kg), T3: 700 ppm PSB	↑Jejunum and SI length, jejunal villi. T2 produced deepest crypts and lowest VH:CD ratio in all intestinal segments at d-14.	[156]
CA	Male Ross 308 broiler for 42 days	Exp. 1: T1: 0, T2: 10, T3: 20, T4: 30 g/kg CA Exp. 2: T1: 0, T2: 30, T3: 60 g/kg CA	Exp.1: ↑ proventriculus weight and IL. ↑ Duodenum, jejunum and ileum, VL. ↑ CD and VL: CD ratio. ↓ epithelial thickness of the Jejunum. Exp. 2: ↑ gizzard weight and IL. ↑ proventriculus, intestine, gizzard, JL and ileum. ↑ VL, CD, and goblet cell count in the hindgut. ↓ Epithelial thickness in the SI.	[126]

Antibiotic as a growth promotor (oxytetracycline, colistin sulfate/Colival) (AB); antibiotic bacitracin methylene di-salicylate (BMD); dextran sulfate sodium (DSS); Negative control (CON); encapsulated SB (consists 99.9% butyrate salt) (ESB); glucose transporter 5 (GLUT5); ileum length (IL); jejunum length (JL); zinc bacitracin (ZnB); peptide transporter (PepT1); partially protected sodium butyrate (PPSB); sheep red blood cells (SRBCs); sodium-dependent glucose transporter (SGLT1); small intestine (SI); uncoated sodium butyrate (UCSB); villi length (VL); villi width (VD) increased (↑); decreased (↓).

## 5. Application of BA and CA in Poultry Nutrition

BA and CA are potent feed additives used in livestock nutrition, and we have gained special insights due to gut health enhancer and antimicrobial activity [21,65,142,145,150]. Physical form, flavor property, water solubility and safety concerns of BA and CA are the crucial factors for effective delivery to target gut compartments of animals including birds [95,157,158]. The rate of acid absorption and the compartments of the GIT are closely associated with the various application techniques. These include unprotected butyrate absorbed in the crew, stomach and gizzard, whereas TB is absorbed in the SI and fat-coated butyrate is absorbed throughout the GI tract [159]. BA can be supplemented as straight (natural BA), butyrate (salt forms), coated/encapsulated (lipid shell) salts of butyrate [57,103] and butyrate glycerides (butyrin) [123]. Each form of the product has its advantages and limitations in terms of bioavailability, cost, handling safety, stability, processing temperature/pressure, releasing target in the GIT, and feasibility [38]. In a comparative study to evaluate encapsulated BA at 500 g/t (T2) and protected SB (PSB) at 700 g/t (T3) of feed in laying hens, the results indicated that T2 (41.3 g/kg BW) showed higher (P = 0.005) weight of SI contents than the control diet (35.0 g/kg BW) and T3 (36.5 g/kg BW) group of birds [160].

In contrast, higher total SCFAs and BA concentration in the cecal digesta were observed in T3 versus T1 and T2 groups. However, the addition of the two BA sources (T2 and T3) increased cecal microflora activity of enzymes in birds and noted beneficial effects on eggshell quality, the tibia, and selected GIT parameters [160]. Similarly, Xiong et al. proved that CSB was less effective than TB in improving gut morphology in LPS-challenged broilers [30] because CSB rapidly absorbs in the upper intestine, whereas TB is slowly reached and is more absorbed in the SI. A supportive study for the above findings was conducted by van den Borne et al. on feeding uncoated vs. fat-coated CB on stomach passage time in the GIT of broilers. The results showed that 80% of uncoated CB is oxidized and absorbed from the upper digestive tract. In comparison, fat-coated CB extended-release pattern time of more than six hours indicated that the coating process delayed release more in the lower intestinal segments [161]. Pires et al. also examined the effects of PSB at 0 or 0.105 g/kg on the percentage of broken and dirty eggs at commercial chicken farms. The number of dirty and broken eggs was reduced in laying hens fed 0.105 g/kg PSB versus the control diet [113]. Likewise, PSB significantly increased carcass weight versus unprotected SB or without butyrate in broilers diet, which indicates the role of butyrate in increasing poultry meat production [84]. Therefore, the above findings showed that targeted delivery of organic acids through a coating or encapsulation process effectively exerts the antimicrobial function in the GIT. This reduces coliform counts both in the distal jejunum, cecum and SI, as the main sites of bacterial activity in chicken [95,162]. There is currently a novel application technique called in ovo administration of BA, that occurs in the early life stages of chickens to establish a healthy intestinal microflora environment in the early post-hatch stage of chicks [163,164]. At the early age of birds, butyrate is used as a direct energy source to the gut epithelium and can reduce intestinal inflammation by promoting mucus secretion [57]. In general, and as a witness for its contribution, the global animal feed industry reported protected BA to be the second top (30%) product as a replacement of antibiotics, followed by probiotics (50%), to improve the gut health of broilers.

Similarly, CA as a potential growth promoter and potent chelator of calcium can produce a favorable environment for endogenous and exogenous microbial enzymes such as phytase, which improves the hydrolysis of phytate available in poultry feedstuffs [17,141]. Fazayeli-Rad et al. showed that 30 g/kg CA improved growth performance and nutrient retention of male broilers at 42 days due to improved phosphorus availability [165]. Islam et al. also studied the effects of dietary CA at 0%, 0.25%, 0.75% and 1.25% on broiler performance and mineral metabolism for 35 days. Bodyweight, feed conversion efficiency (FCE), carcass weight, and graded carcasses increased sequentially with increasing CA levels. In addition, mineral digestibility, bone ash content and mineral density, and strength also significantly increased up to 0.75% level of CA [125]. A supportive study on growing rabbits at 0%, 0.5%, 1.0% and 1.5% CA supplementation was examined for 56 days. The growth rate and growth velocity increased linearly with the increasing CA addition and concluded that 1.5% was an effective dosage for rabbits’ positive growth performance [166]. In summary, the type of product, inclusion level, bird type and age, diet composition and particle size, methodology, experimental duration, LPS challenges, and environmental stress are the main factors determining the effectiveness of BA and CA on broiler performance and intestinal development (Table 1 and Table 2).

## 6. Conclusions and Future Research Directions

The goal of market-oriented modern poultry production depends on high feed efficiency and the health of the gut. The withdrawal of antibiotics in the diet of poultry brought numerous options as antibiotics alternatives, among which organic acids acquire significant attention. In animal nutrition, organic acids and their salts are cost-effective, performance-enhancing options and exert antibacterial, pH reduction effects with the function of energy supply. BA and CAs, as representatives of organic acids, significantly impact their biological functions of antimicrobial activity as a gut health enhancer. They are used as the main energy source of metabolic activities and reduce pathogenic bacterial load in the GIT of birds. The supplementation results in poultry are not consistent due to different conditions such as the type of product (application methods), inclusion levels, target release of the products in the GIT, age, and sex strains of birds. BA and CA are promising antibiotic replacers and are significant concerns in future research work on poultry nutrition. In addition, blends of organic acids or simple monocarboxylic acids/SCFA (e.g., BA) and Krebs cycle acids/TCA/carboxylic acids bearing a hydroxyl group (e.g., CA) are not common and need further investigation about their interactive effects on intestinal microbiota composition, diversity, gut health, and immunity as well as growth performance in poultry.

## Figures and Tables

**Figure 1 ijms-22-10392-f001:**
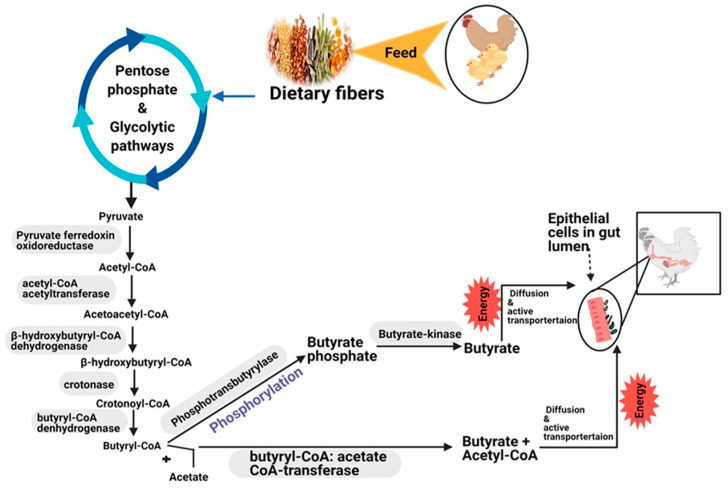
Production pathways and absorption mechanism of butyrate (butyrate metabolism) from dietary fibers in the intestine of monogastric animals.

**Table 1 ijms-22-10392-t001:** Effects of the different forms of BA and CA on growth, digestibility and carcass yield of broilers.

Forms of BA/CA	Broiler Strain and Trial Duration (Day)	Study Layout and Dosage Levels	Responses Expressed as a Percentage of Respective Controls	Reference
EBA (Buti PEARL)	Male Cobb broilers for 42 days	Exp. 1: T1: CTR, T2: 100, T3: 200 and T4: 300 g/t EBA respectively. Exp. 2: Similar with Exp.1 but added T5:400 and T6: 500 g/t EBA.	EXP 1: ↓ FC at d 0–21. ↑ BWG at d-35 and d-42. EXP 2: ↓ FC at T6. ↑ BWG at d-35 and 42 inT4, T5 and T6.	[120]
MEBA	Hubbard classic	T1: CTR, T2: 0.25, T3: 0.35 and T4: 0.45 g/kg of MEB.	↑ BWG, FCR and AID with addition of MEB.	[121]
TB and FCSB	Ross 308 Broilers for 35 days	T1: CTR, T2: TB (53% BA), T3: FCSB (24% BA)	At d-25 and 35, ↑BWG at T2 (0.058 kg) and T3 (0.043 kg). At d 9–25, ↑FCR at T2 by 5 points, T3 by 6 points. T2 and T3 ↑ FCR in all periods by 4 and 5 points respectively.	[122]
MB and TB	Ross 308 male broilers for 42 days	Exp. 1: T1: CTR, T2: 500, T3: 1000, T4: 2000, T5: 3000 ppm MB. Exp. 2: T1: CTR, T2: 5T5M, T3: 5T5Ms, T4: 5T20M, T5: 5T20Ms	Exp. 1: ↓abdominal fat deposition. ↑Breast muscle. Exp. 2: ↑Breast muscle weight in T2 at 5 weeks of age.	[123]
BA	Commercial broilers for 35 days	T1: CTR, T2: Antibiotic (furazolidone), T3: 0.2%, T4: 0.4%, T5: 0.6% BA	↑ FCR, dressing % and ↓ abdominal fat content	[15]
BA, CA	Unsexed Ros 308 broiler for d-42	3 × 3 factorial CRD: CP levels (H, M, L) and 3 dietary OA (CTR, 2.5 g/kg CA or BA)	M + L CP ↓ ADG at d 0–14 and d 14–28. CA ↑ ADG at d 0–14. Both CA + BA ↑ADG, FCR and carcass yield, ↓ gizzard weight at d-42.	[117]
CA	Vencob broilers for 42 days	T1: CTR, T2: 2.4, T3: 3.2, T4: 4.00 mg/kg CA respectively	↑ FCE better in T3 followed by T2, T1 and T4. ↑dressing% and carcass yield in T3.	[124]
CA	Ross broiler chicks for 35 days	T1: CTR, T2: 0.25, T3: 0.75%, T4: 1.25% CA	↑ BWG, FCE, microminerals digestibility, bone ash and mineral density, and strength at T3. ↑ Slaughter weight and carcass quality with CA addition.	[125]
CA	Male Ross 308 broiler for 42 days	Exp. 1: T1: CTR, T2: 10, T3: 20, T4: 30 g/kg CA Exp. 2: T1: CTR, T2: 30, T3: 60 g/kg CA	Exp.1: ↑ ADG, FCR and nutrient digestibility. Exp. 2: ↑ ADG and ADFI. ↑ ICPD, AME and tP.	[126]

Apparent ileal digestibility (AID); Apparent metabolizable energy (AME); Average daily feed intake (ADFI); Body weight gain (BWG); crude protein (CP); control (CTR); encapsulated BA (Buti PEARL) (EBA); fat-coated sodium butyrate (FCSB); High (H); Ileal crude protein digestibility (ICPD); low (L); medium (M); microencapsulated BA (MEBA); organic acid (OA); Tributyrin (TB); total phosphorus (tP); 5T5M = 500 ppm TB + 500 ppm MB; 5T5Ms = 500 ppm TB + 500 ppm MB staggered; 5T20M = 500 ppm TB + 2000 ppm MB; 5T20Ms = 500 ppm tributyrin + 2000 ppm MB staggered; increased (↑); decreased (↓).

## Data Availability

Not applicable.

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
