# Peer review of "Butyric and Citric Acids and Their Salts in Poultry Nutrition: Effects on Gut Health and Intestinal Microbiota"

_ijms, 2021, doi:10.3390/ijms221910392_

Round 1

Reviewer 1 Report

The review could be interesting and it is well planned, but it contains such a number of errors that it should be profoundly modified.

Almost all the references in Tables 3 and 4 are wrong. They are highlighted papers in the manuscript and these types of errors are not acceptable.

In addition, I have other considerations. For example, the title is very broad and includes “intestinal microbiota”. In my opinion, there is no in-depth review of the changes these acids exert on the gut microbiota communities from broilers.

In addition, throughout the manuscript there are some other mistakes (such as writing some names of bacteria in small letters) and many references are not correct. Possibly the authors have had some problem when including the references.

Table 2

Ref 92. the reference is not correct. I think that correct is the ref 109

Ref 93. the reference is not correct.

Ref 94. Paper about Lactobacillus, the reference is not correct

Ref 95. the reference is not correct

Etc.

Table 3

Ref 25 the paper is about Salmonella but the results do not correspond to what is cited in the text, I think the reference is not correct.

Ref 122, assay in M77 Hubbard broiler at d-21 but no corresponde with cited paper.

Ref 123. the reference is not correct.

Ref 104. The authors highlight a study conducted in Scophthalmus Maximus (fish). Broilers and fish? I think that reference is not correct.

Author Response

We thank you very much for your fruitful comments and suggestions. Please see the attachment.

Reviewer 2 Report

Melaku et al., reported that butyric and citric acids affect on gut health and intestinal microbiota in poultry. This review is helpful for those who want to get these knowledges for the health in poultry.

Minor comments

Please check carefully about bacterial scientific name.

Line 61,  “ salmonella typhimurium

“clostridium perfringens”

“salmonella enteritidis“    

Salmonella typhimurium

Clostridium perfringens

Salmonella enterica subsp. enterica serovar Enteritidis  

Line 155, Campylobacter,    and  Shigella

genus Campylobacter or Campylobacter sp.

genus Shigella or Shigella sp.

Line 187, Eimeria species

Eimeria species

              (non-italic)

Line 85 prefer “citric acid “

Line 179, coccidiosis

Line 188, Broilers fed MESB challenged with Necrotic Enteritis (NE) showed….

      What was challenged?

Line 224, prefer ” leaky gut” ragher than “gut leakage “

Author Response

(The authors gave the same response as above.)

Reviewer 3 Report

Review of the manuscript ID: ijms-1321178, titled “Butyric and citric acids and their salts in poultry nutrition: Effects on gut health and intestinal microbiota”.

The authors of the manuscript presented the current knowledge on the use of selected organic acids in poultry nutrition and they performed this task without any problems.

The subject matter discussed in the work is particularly important taking into account the withdrawal of feed antibiotics as antibacterial growth promoters and is in line with the current and still open trend regarding the use of safe functional feed additives in animal nutrition.

The authors' knowledge is based on over 150 items of literature, the vast majority of which are the most recent. I believe that this is a valuable and substantive publication.

The work requires a slight correction and systematization of the presented items of references.

The review of studies presented in the manuscript should be supplemented with information on the negative effects (impact on health and production effects) of excessive doses of organic acids used in animal nutrition.

Throughout the manuscript repeated errors improper use reference numbers; for example:

Line 291: There is [103] and it should be [121]

Line 297: There is [25] and it should be [29]

Line 280: an invalid reference to reference number [61]

Line 288: an invalid reference to reference number [117]

Line 310: an invalid reference to reference number [119]

Line 319: There is [125] and it should be [142]

Line 392: There is [140] and it should be [157]

Author Response

(The authors gave the same response as above.)

Reviewer 4 Report

Dear authors,

please find my comments and suggestions in the attached pdf.

Author Response

(The authors gave the same response as above.)

Round 2

Reviewer 1 Report

I have read your work carefully and have seen the improvements made to the manuscript after the first review. I think the manuscript is interesting and provides a good idea of the benefits of butyric and citric acids and their salts in poultry nutrition. 

Author Response

Dear reviewer;

We thank you very much from the bottom of our heart for your careful review and fruitful comments you gave us.

With great respect!

Reviewer 4 Report

Dear Authors

Many thanks, however I oberved that there are still many mistakes grammatically as well as scientifically and needed to be improved according to my last comments. Please do them critically.

Maybe the paper could be reconsidered after doing the comments critically.

Author Response

Dear Reviewer;

We thank you very much for all your careful review.

We have done all the necessary comments as shown from the attachment.

All the red color track mode in the manuscript are your comments corrected. The green font color are the comments given by the other reviewers and corrected in the first round by the authors.

Many thanks

Great respect
